# Estimation of the Local Concentration of the Markedly Dense Hydroxyl Radical Generation Induced by X-rays in Water

**DOI:** 10.3390/molecules27030592

**Published:** 2022-01-18

**Authors:** Ken-ichiro Matsumoto, Megumi Ueno, Yoshimi Shoji, Ikuo Nakanishi

**Affiliations:** 1Quantitative RedOx Sensing Group, Department of Radiation Regulatory Science Research, National Institute of Radiological Sciences, Quantum Life and Medical Science Directorate, National Institutes for Quantum Science and Technology, 4-9-1 Anagawa, Inage-ku, Chiba-shi 263-8555, Chiba, Japan; ueno.megumi@qst.go.jp (M.U.); shoji.yoshimi@qst.go.jp (Y.S.); 2Quantum RedOx Chemistry Group, Institute for Quantum Life Science, Quantum Life and Medical Science Directorate, National Institutes for Quantum Science and Technology, 4-9-1 Anagawa, Inage-ku, Chiba-shi 263-8555, Chiba, Japan; nakanishi.ikuo@qst.go.jp

**Keywords:** hydroxyl radical, X-ray, ionizing radiation, electron paramagnetic resonance, spin-trapping, DMPO, water radiolysis, local concentration, intermolecular distance, linear-density

## Abstract

The linear-density (number of molecules on an arbitrary distance) of X-ray-induced markedly dense hydroxyl radicals (•OH) in water was estimated based on EPR spin-trapping measurement. A lower (0.13 mM–2.3 M) concentration series of DMPO water solutions and higher (1.7–6.0 M) concentration series of DMPO water solutions plus neat DMPO liquid (8.8 M as DMPO) were irradiated with 32 Gy of X-rays. Then, the yield of DMPO-OH in DMPO water solutions and the total spin-adduct of DMPO in neat DMPO were quantified. For the higher concentration DMPO series, the EPR peak area was estimated by double integration, and the baseline correction of the integral spectrum is necessary for accurate estimation of the peak area. The preparation of a suitable standard sample corresponding to the electric permittivity according to DMPO concentration was quite important for quantification of DMPO-OH, especially in DMPO concentration beyond 2 M. The linear-density of •OH generation in water by X-ray irradiation was estimated from the inflection point on the plot of the DMPO-OH yield versus DMPO linear-density. The linear-density of X-ray-induced markedly dense •OH was estimated as 1168 μm^−1^, which was converted to 0.86 nm as the intermolecular distance and 2.6 M as the local concentration.

## 1. Introduction

Ionizing radiation, even electromagnetic waves (photons) or particles, can ionize and/or excite molecules in the material transmitting the ionizing radiation. When the ionizing radiation is traveling through water, the water molecules may be ionized/excited and decomposed to make initial reactive species such as hydroxyl radicals (•OH), hydrogen radicals (•H), and hydrated electrons (e^−^_aq_) [1,2,3,4]. This is termed water radiolysis. Sequentially, the secondary reactive species, such as hydrogen peroxide (H_2_O_2_), hydroperoxyl radical (HO_2_^•^), superoxide (O_2_^•−^), etc., can be generated through the reactions of initial reactive species and/or through the reaction of initial species with oxygen [5,6,7].

The spur diffusion model is well established as a reaction process of initial reactive species generated by the radiolysis of water [8,9,10,11,12]. The chemical process of radiolysis of water has been analyzed based on pulse radiolysis. Pulse radiolysis observes the absorbance of target chemical species over time after microsecond or shorter pulsed irradiation, and analyzes the dynamics of the target species. By ionization or excitation of water molecules, some initial reactive species, i.e., ions and radicals, can be generated and distributed in a volume of ~1 nm in diameter, which is termed a “spur” [10]. For photon radiation, spurs were thought to be scattered and well separated with an internal distance of hundreds of nm [11,12]. The reactive species generated in the spur react with each other and diffuse to be uniformly distributed during ~10^−6^ s [13,14].

Local ionization and free radical formation on biological molecules, such as DNA, sugar, and protein, have been investigated using electron paramagnetic resonance (EPR) spectroscopic techniques and well summarized in a book [15,16]. Direct and indirect free radical formations on DNA and its constituents after irradiation of photon and/or charged particle beams were investigated using EPR spectroscopy [15]. Using the EPR spin trapping technique, radiation-induced free radical species on DNA constituents and/or proteins have been identified [16]. On the other hand, EPR investigation of local free radical generation by radiation in a liquid state, such as in water or lipids, is in progress due to the instability of generated free radicals even using spin trapping techniques.

•OH is recognized as the most reactive chemical specie among the radiation-induced reactive oxygen species (ROS) [17,18,19]. The initial distribution and concentration of •OH is important for predicting chemical reactions and biological effects of ROS. Initial yields of •OH were estimated from the picosecond and/or sub-picosecond time course of initial species measured by pulse radiolysis, and reported to be 5.6–5.9 mol/100 eV [20,21,22], which is the so-called initial G-value of •OH.

The EPR spin-trapping technique is used for the chemical analysis of short-lived free radicals. Recently, two different appearances of •OH generation in aqueous solutions irradiated by ionizing radiation, either X-rays or carbon-ion beams, were observed using a method based on EPR spin-trapping [23,24,25,26]. Those are relatively sparse and markedly dense •OH generation. The relatively sparse •OH generation yields a local concentration of mmol/L, whereas the markedly dense •OH generation yields a local concentration of mol/L. The ratio of the two different appearances of •OH generation varied by different linear energy transfer (LET) of carbon-ion beam, and the yields of sparse •OH decreased with increasing LET [23,25]. The molecular distances of •OH estimated by the EPR spin-trapping method were not consistent with the spur model; however, the ratio of sparse •OH generation decreased with increasing LET.

The local concentration of •OH generation is highly important to decide the reaction that follows, especially for the generation of H_2_O_2_. The markedly dense •OH generation in water may be related to O_2_-independent H_2_O_2_ generation [26,27]. O_2_-independent H_2_O_2_ generation may play important roles in low pO_2_ conditions in vivo. The H_2_O_2_ generation under hypoxic conditions likely results from the direct reaction between •OH, and two or more •OH must be generated in a very close area to react with each other [26,27]. On the other hand, the sparsely generated •OH cannot react due to their distance, which was estimated to be 4–7 nm in a previous report [25]. The intermolecular distance between the markedly dense •OH is must be less than 0.8 nm [26]; however, no concrete numerical inter-molecular distance of the markedly dense •OH has been reported. These differences in ROS generation are a factor determining the quality of radiation [28].

The distribution of radiation-induced free radical species is not uniform in the 3-dimensional space because ionization is a sequential event occurring on the track of photons, electrons, or charged particles. The conventional concept of density and/or concentration, which indicates weight/number of molecules per volume, is insufficient to represent the amount of eccentrically located molecules (Appendix A). Therefore, the description of 1-dimensional molecular density, i.e., the number of molecular probes in the unit of distance, was required to estimate the local distribution of the radiation-induced reactive species. This is now termed “linear-density” in this manuscript, and the concept of linear-density was again introduced and defined. The linear-density, which is defined as the number of molecules aligned on the linear unit of distance, is the reciprocal of the inter-molecular distance of the subjected molecules at a certain concentration.

•OH can be measured by an EPR spin-trapping method using 5,5-dimethyl-1-pyrroline-*N*-oxide (DMPO) as the spin-trapping agent (Figure 1). Using a series of several concentrations of DMPO solutions, DMPO-OH induced in each DMPO concentration was quantified and plotted versus the DMPO linear-density (μm^−1^). The plot of DMPO-OH versus DMPO linear-density exhibits a characteristic 3-phase profile. The yield of DMPO-OH in the irradiated DMPO solution increased with DMPO linear-density (phase 1). When the linear-density of DMPO was higher than the linear-density of •OH generated in the sample solution, the yield of DMPO-OH became constant and phase 2 was observed. The inflection point created by phases 1 and 2 gives the linear-density of relatively sparse •OH generation. The appearance of the 3rd phase reflects another appearance of •OH generation at a markedly high linear-density. Phase 3 may continue to increase, but the end of phase 3 was not observed in previous experiments [23,24,25,26]. Phase 4 should start when the linear-density of DMPO in the sample solution is higher than the linear-density of markedly dense •OH generation, and the yield of DMPO-OH in the irradiated sample solution should become constant. However, there have been no data for DMPO solutions more concentrated than 2.3 M due to technical difficulties until now. Accurate quantification of DMPO-OH in the high concentration DMPO solution is difficult due to the change in the EPR detection sensitivity associated with the expected change in dielectric nature of the sample solution. The change in the EPR spectral shape will also interrupt the quantification process of DMPO-OH.

The purpose of this study was to improve the procedure for quantifying •OH based on the yield of DMPO-OH and estimate the local linear-density of X-ray induced markedly dense •OH generation in water.

## 2. Results

Using a series of DMPO reaction mixtures containing varying concentrations (0.13 mM–2.3 M), the linear-density of •OH generation was estimated. The time course of single EPR line data (2nd line of DMPO-OH) was analyzed, and the C_0NET_ value was obtained as described in the Materials and Methods. A plot of C_0NET_ values of DMPO-OH versus linear-density of DMPO shown in Figure 2 (circles) showed a typical 3-phase profile. As described in the figure legend, the linear-density of relatively sparse •OH generation was 142 μm^−1^, which was converted to 7.0 nm as the intermolecular distance and 4.8 mM as the local concentration. The yield of the sparse •OH, i.e., the value on the *y*-axis at the inflection point of phases 1 and 2, was estimated as 194 nmol/L/Gy (horizontal gray dotted arrow). The highest value in phase 3 estimated in the experiment using the lower concentration series of DMPO was 501 ± 36 nmol/L/Gy, which was the lower estimation of total •OH generation.

To estimate the rest of phase 3, the higher concentration (1.7–6.0 M and neat) DMPO water solutions were prepared and X-ray-induced spin-adducts in the solutions were measured. An EPR spectrum observed in the DMPO water solutions and the neat DMPO irradiated with 32 Gy of X-rays is shown in Figure 3. The EPR spectrum observed in the 1.7 M DMPO solution is shown in Figure 3A, in which typical 1:2:2:1 spectral shapes of DMPO-OH and overlapping DMPO-H can be distinguished. An EPR spectrum obtained in the 2.3 M DMPO solution, in which the linewidth of both DMPO-OH and DMPO-H is slightly broader than that in the 1.7 M DMPO solution, is shown in Figure 3B. EPR spectra obtained in the 3.2, 4.8 M, and 6.0 M DMPO solution are shown in Figure 3C–E. In the 3.2, 4.8 M, and 6.0 M DMPO solution, the spectrum of DMPO-OH exhibited extra hfs due to the differing solvent dielectric nature, i.e., changing from water to DMPO. EPR spectrum of DMPO-OH is composed of 3 of 1:1 splitting. The 1:1 splitting was made by 1 hydrogen and again split into three by nitrogen. In water, hydrogen and nitrogen give the same splitting width and a 1:2:2:1 pattern. Splitting by hydrogen was narrower in low dielectricity solvent, and then changed to six lines. In higher DMPO concentrations over 6.0 M, the 6-line EPR spectrum was observed clearly (Figure 3E). An EPR spectrum obtained in the neat DMPO liquid, whose purity was 99% and corresponded to a concentration of 8.8 M, is shown in Figure 3F. The spectrum in the neat DMPO may be consistent with that in DMPO-OH and DMPO-OR (alkoxyl radical adduct) [29].

The EPR spectral shape was altered by the dielectric nature of the solvent in the higher concentration DMPO solutions. Therefore, the intrinsic EPR sensitivity depending on the material also changed. For quantification of free radical adducts observed in the sample DMPO solutions, the standard solution of TEMPOL must prepared with the same solvent, i.e., same dielectric nature, as the sample solution. The EPR signal intensities of 0.1 mM TEMPOL solutions prepared using 1.7, 2.3, 3.4, 4.8, or 6.0 M DMPO water solution or neat DMPO liquid as the solvent were compared (Figure 4). Signal intensities of 0.1 mM TEMPOL dissolved in several different DMPO concentrations were almost similar until 2.3 M DMPO solution; however, a higher signal intensity was observed when the concentration of DMPO exceeded 2.3 M. The signal intensity of 0.1 mM TEMPOL in 4.8 and 6.0 M DMPO 1.6 and 1.8-times higher than that in 0.1 mM TEMPOL in water. The signal intensity of 0.1 mM TEMPOL in neat DMPO (8.8 M as DMPO) was 1.6-times higher than in water. For the experiments using the lower concentration series of DMPO, a 1 mM water solution of TEMPOL was used as a standard for all samples measured. On the other hand, for the experiments using the higher concentration series of DMPO, 0.1 mM TEMPOL standard solutions were prepared using the same solvent of each sample. Thus, every sample solution was used as a solvent for its standard TEMPOL solution.

The estimated net concentration of DMPO-OH in the irradiated DMPO solutions or total spin-adducts in the irradiated neat DMPO were plotted versus the linear-density of DMPO in the sample solution (Figure 2). The results of the experiments using a lower and higher concentration series of DMPO overlapped, as shown in Figure 2. The circles and triangles indicate the results of the experiments using the lower and higher concentration series, respectively. The quantified values for DMPO-OH yield in 1.7 and 2.3 M DMPO solutions, common samples in both experiments, were almost the same in both experiments. The results of the neat DMPO experiment are indicated by a closed square and the total spin amount was plotted just for reference. The experiment using the higher concentration series of DMPO yielded the 4th phase (open triangles). The linear-density of markedly dense •OH generation was able to be estimated from the inflection point of the plot profile, and the linear-density was 1168 μm^−1^, which was converted to 0.86 nm as the intermolecular distance and 2.6 M as the local concentration. The *y*-axis value of the inflection point to phases 3 and 4, which corresponds to the total •OH yield, was 529 nmol/L/Gy. The calculated yield of •OH using the initial G-value of 5.6 n/100 eV [22] was 581 nmol/L/Gy. Therefore, this method may detect 91% of the initial •OH yield. However, there is a concern that a part of DMPO-OH observed in aerobic condition might be originated from direct ionization of DMPO, and •OH generated under aerobic condition might be slightly overestimated. It is expected that the fact would be clarified by an experiment under hypoxic condition, which is planned but some technical improvement would be required for keeping hypoxic condition through irradiation to end of EPR measurement.

The component of phases 1 and 2, which was the result of trapping relatively sparse •OH, overlapped with the component of phases 3 and 4, which was the result of trapping markedly dense •OH. If the markedly dense •OH generation can be attenuated, phase 2 should become almost horizontal. However, the experimentally observed phase 2 was gradually increased by the phase 3 component when the linear-density of DMPO increased. The detailed concept of the local density estimation was described and the structure of the 3-phase profile was simulated in a previous paper [26].

In this study, the point at 1000 µm^−1^ on the horizontal axis was empirically predicted as a component of phase 3 based on results of our previous reports [23,24,26]. The point at 1000 µm^−1^ remained clearly at phase 3, when LET was increased [23] and when the sparse •OH component, i.e., phases 1 and 2, was canceled by a scavenger [24,26].

X-ray-induced DMPO-spin-adducts observed in neat DMPO included DMPO-OH and DMPO-OR [29]. The DMPO-OH and DMPO-OR in neat DMPO may be generated by spin-trapping free radicals on a DMPO molecule, which should be caused by direct ionization of DMPO molecules. There may be no •OH from water radiolysis observed in this case. Therefore, the total amount of DMPO-spin-adducts in neat DMPO caused by X-ray irradiation was measured and the net yields considering its decay during irradiation was estimated. The value obtained for neat DMPO (black square in Figure 2), which is considered the total yield of DMPO-OH and DMPO-OR [29], is only a reference and was not used to obtain phase 4. However, the value obtained for neat DMPO may indicate the amount of total ionization.

## 3. Discussion

DMPO is the most popular spin-trapping agent for EPR detection, identification, and/or quantification of unstable free radical species in water. The structure of DMPO is shown in Figure 1, having a molecular weight of 113.16. The molar concentration of the neat 99% liquid DMPO was calculated to be 8.8 M. In the 8.8 M neat liquid DMPO, a single DMPO molecule occupies a volume of 0.19 nm^3^. The diameter of a 0.19 nm^3^ spherical volume can be calculated as 0.71 nm or the edge length of a 0.19 nm^3^ cube can be calculated as 0.57 nm, which can be considered the intermolecular distance of DMPO. In this study, the edge length of cubic volume was indicated as the intermolecular distance.

For accurate quantification, a calculated Gaussian line shape was fitted onto one isolated single isotropic line of the experimentally observed 4-line EPR spectrum of DMPO-OH (Figure 5), and then the linewidth and spectral height of the fitted Gaussian line was obtained to calculate the area of integrated spectrum in the previous reports using in-house software [23,24,25,26]. Observation of an isolated single isotropic line is difficult in a high-concentration DMPO solution due to the appearance of additional hyper fine splitting (hfs) associated with changes in the polarity of the sample solution, in addition to distortion of the line shape or line broadening associated with anisotropy caused by higher viscosity of the sample solution. Double integration of digital spectrum data may be effective only for data with a sufficiently low noise level because noisy data cause variability in the double integration value due to distortion of the baseline.

The double integration value, i.e., area, of the EPR spectrum is proportional to the amount of spins in a certain volume of the sample. Distortion of the spectrum due to broad noise was a problem in the area estimation for quantifying free radicals using EPR. The line fitting method can ignore baseline distortion by fitting a distortion-free calculated line shape on an experimentally observed spectrum. However, the line fitting method is only available if a well-separated single isotropic line is observed. For multiple overlapping spectra or a spectrum having incompletely split narrow hfs, the double integration method is the standard manipulation for area estimation. However, distortion of the baseline on the integrated spectrum yields a relatively large positive or negative area, as shown in Figure 6. In this study, the distortions of the integrated spectra were manually corrected.

Using the higher concentration series of DMPO, the net concentration of DMPO-OH in the irradiated sample solutions was estimated. However, the EPR spectral shape of DMPO-OH in a highly concentrated DMPO solution may have extra hfs and no simple single Gaussian line can be obtained on the spectrum. Therefore, area estimation of the integrated spectrum is required to quantify spin amount. In this case, the area of integral spectrum, i.e., the double integrated value of the EPR spectrum, included both DMPO-OH and DMPO-H in DMPO water solutions or total spin-adducts of DMPO in neat DMPO. To avoid the contribution of DMPO-H, the area of the center peak of DMPO-H was separately observed and the 6-fold value of its area was subtracted from the total area (Figure 6E). DMPO-H in water gives a 9-line EPR spectrum with intensity of 1:1:2:1:2:1:2:1:1 [17]. Therefore, the center line has 2/12 = 1/6 of the total intensity.

An example of the efficiency of this baseline correction is shown in Figure 7. The baseline correction of the integrated spectrum greatly improved the reproducibility of the different experiments as shown in Figure 7B.

In 1.6, 2.3, 3.4, 4.8, 6.0 M DMPO water solutions, one DMPO molecule existed every 1.00, 0.73, 0.51, 0.34, or 0.28 nm^3^ volume, and the intermolecular distance of DMPO was 1.0, 0.9, 0.8, 0.7, or 0.65 nm, respectively. Space between a DMPO molecule and the other DMPO molecule was filled with water molecules. Water is a small molecule consisting of two hydrogen atoms and an oxygen atom. The molar concentration of pure water is 55.5 M, in which a single water molecule occupies 0.030 nm^3^ and the intermolecular distance is 0.31 nm. In 1.6, 2.3, 3.4, 4.8, 6.0 M DMPO water solutions, the number of water molecules per one DMPO molecule are 27.2, 18.1, 10.9, 5.2, or 3.0, respectively.

In the 6.0 M DMPO solution, one DMPO molecule was monitoring only three vicinal water molecules, and one of the three water molecules was ionized/excited to give •OH; therefore, the trapping efficiency of the DMPO for this initial •OH molecule may be greater than 90%. Even in 6.0 M DMPO solution, the DMPO-OH adduct was mainly generated by X-ray irradiation instead of the DMPO-OR adduct, as shown in Figure 3E. The DMPO-OR adduct may be generated by scavenging a carbon-centered radical generated from a directly ionized DMPO molecule. In addition, in the 6.0 M DMPO solution, the DMPO-H adduct was generated by X-ray irradiation, as shown in Figure 3E. Therefore, the DMPO likely trapped •OH and •H generated from water even in the 6.0 M DMPO solution. On the other hand, both DMPO-OR and DMPO-OH adducts were observed in neat DMPO irradiated by X-rays, as reported in a previous paper [29].

In 6.0 M DMPO solution, the molecular distance of DMPO is 0.65 nm and the molecular distance of water is 0.45 nm, corresponding to 17.9 M as the concentration of water molecules. The initial photon or secondary electron passing through this solution can therefore meet DMPO every 0.65 nm and a water molecule every 0.45 nm, and ionize them. It is unclear how much ionization occurs with one DMPO molecule. However, the DMPO molecule trapping the •OH is not necessary on the track. When both DMPO and water existing on the track are ionized simultaneously, and a carbon center radical on the DMPO and •OH from water is induced, a DMPO molecule nearby may react with •OH due to the reactivity of •OH.

The EPR spin-trapping technique is a type of destructive measurement technique used to investigate water radiolysis. The EPR spin-trapping technique can detect one or two free radical species generated through the water radiolysis process during the life time of the radical. As the trigger-free radical of the sequential reaction must be trapped in order to be detected, the time course of the sequential reactions induced by the target radical cannot be followed. The ionizing radiation can interact with water molecules, ionizing or exciting them. The ionization of water (Equation (1)) can give e^−^_aq_ (Equation (2)), •OH and H^+^ (Equation (3)), and •H (Equation (4)). The excitation of water can give •OH and •H (Equation (5)).
X-ray ~~~> H_2_O → H_2_O^+^ + e^−^(1)
e^−^ + *n*H_2_O → e^−^_aq_(2)
H_2_O^+^ → •OH + H^+^(3)
H^+^ + e^−^_aq_ → •H(4)
X-ray ~~~> H_2_O → H_2_O* → •OH + •H(5)

DMPO can scavenge •OH, •H, and e^−^_aq_, and can repress the reactions following (Equations (6)–(9)). EPR spin trapping detection may more likely disrupt the spur diffusion process of initial reactive species.
•H + O_2_ → HO_2_^•^(6)
e^−^_aq_ + O_2_ → O_2_^•^^−^(7)
HO_2_^•^ + HO_2_^•^ → H_2_O_2_ + O_2_(8)
HO_2_^•^ ⇄ O_2_^•^^−^ + H^+^(9)

When •H and/or e^−^_aq_ were trapped by the spin-trapping agent, HO_2_^•^/O_2_^•^^−^ was not sufficiently generated. Indeed, during DMPO or CYPMPO spin-trapping measurement of radiation-induced free radicals, -OH and -H adducts of the spin-trapping agent were observed, but no traces of -OOH adducts were found [17,30,31]. The time window for the spin-trapping method naturally depends on the lifetime of the target radical. Therefore, this method should be dedicated to •OH or part of •H during a nanosecond time window. Thus, the effects of HO_2_^•^ or O_2_^•^^−^ on detecting X-ray-induced •OH and •H by DMPO were almost negligible [31].

In this study, the lower concertation series of DMPO was prepared using phosphate buffer and the higher concentration series of DMPO was prepared using milli-Q water. The buffered DMPO solutions were always pH 7.0, whereas DMPO solutions prepared by milli-Q water were pH 5.5–6.0, which was confirmed by pH test stick with a pH 0.5-step colorimetric standard. It was previously demonstrated that the DMPO-OH yield in both DMPO solutions prepared by phosphate buffer and milli-Q water was almost the same [32].

The experiment was performed under aerobic conditions and the DMPO solutions were equilibrated in air. The oxygen concentration in the sample solutions measured by EPR oximetry using lithium-phthalocyanine as the oxygen probe [33,34] was slightly higher (only 1.01-fold higher than that of PB) in 6.0 M DMPO solution than in the other concentrations tested at room temperature (20–22 °C). The O_2_ and CO_2_ concentrations dissolved in water equilibrated in air were predicted to be 0.28 mM and 0.016 mM or less under room temperature (20 °C), when the CO_2_ concentration in air was assumed to be 420 ppm. The CO_2_ concentration in the reaction mixture was sufficiently low compared with the local concentrations of ionization, which was several mM or M. The effects of dissolved CO_2_ are therefore negligible under natural aerobic conditions. In addition, CO_2_ was more soluble in water than DMPO. The estimated CO_2_ solubility in water and 6.0 M DMPO was 1.76 ± 0.16 mg/mL and 1.50 ± 0.16 mg/mL, respectively.

Based on this study, •OH generated less than 0.43 nm from a DMPO molecule, i.e., •OH generated within a 0.86-nm range with DMPO as the center, was trapped and observed. The second •OH generated simultaneously at the same locale may not be accounted for; however, such sequential generation of •OH less than 0.86 nm away may be rare. In addition, the total spin amount caused in neat DMPO was almost similar to the DMPO-OH yield observed in 3.2–6.0 M DMPO solutions. Therefore the frequency of ionization in water and neat DMPO was likely the same. Thus, the dense ionization may have been caused every 0.86 nm or closer, whereas sparse ionization occurred every 7 nm or closer in water and DMPO.

Markedly dense •OH generation was observed experimentally using the EPR spin-trapping technique, which will be an important foothold for future investigations of radiation-induced free radical reactions.

## 4. Materials and Methods

### 4.1. Chemicals

A 99% purified product of DMPO was purchased from Dojindo Laboratories, Ltd. (Kumamoto, Japan). The 99% DMPO is a clear liquid with slight viscosity and specific gravity of 1.01 (Material Safety Data Sheet, Dojindo). 4-Hydroxyl-2,2,6,6-tetramethylpiperidine-*N*-oxyl (TEMPOL) was purchased from Sigma-Aldrich (St. Louis, MO, USA). Other chemicals were of analytical grade. Deionized water (deionization by the Milli-Q system, Merck Millipore, Billerica, MA, USA) was used for all experiments.

### 4.2. Sample Preparation for Estimating the Linear-Density of •OH Generated in a Water Sample Irradiated by X-rays

A series of reaction mixtures containing varying concentrations (0.13 mM–2.3 M) of DMPO was prepared using 100 mM phosphate buffer (pH 7.0) containing 0.05 mM DTPA as the solvent. Another series of DMPO solutions (1.7–6.0 M) dissolved in milli-Q water and neat DMPO liquid (8.8 M as DMPO) was prepared. The DMPO solution or neat DMPO liquid was then transferred to a micro tube.

### 4.3. Irradiation by X-rays

The sample solutions were irradiated with 32 Gy of X-rays using the conditions described below and then used for EPR measurement as soon as possible after irradiation. X-ray irradiation was performed with PANTAK 320S (Shimadzu, Kyoto, Japan). The effective energy was 80 keV under the following conditions: the X-ray tube voltage was 200 kV, X-ray tube current was 20 mA, and the thickness and materials of the pre-filter were 0.5-mm copper and 0.5-mm aluminum, respectively. The dose rate of X-ray irradiation was 3.0–3.1 Gy/min when the distance between the X-ray tube and the sample was 280 mm. Gy means the dose that can be measured physically via the ion chamber.

### 4.4. EPR Measurement of Sample Solutions Irradiated by X-rays

The sample irradiated was immediately used for EPR measurement after irradiation. One hundred microliters of the reaction mixture was drawn up into 25-cm PTFE tubing (i.d. 0.32 ± 0.001 inches, wall 0.002 ± 0.0005 inches; ZEUS, Orangeburg, SC, USA). The PTFE tubing containing sample solution was placed in the TE mode cavity using a special sample holder and measured by an X-band EPR spectrometer (JEOL, Tokyo, Japan). Then, the time course of EPR signal of DMPO-OH or DMPO adducts in neat DMPO was measured. The second peak from the lower field of the 4-line spectrum was recorded for the lower DMPO concentration series (0.13 mM–2.3 M in phosphate buffer) and the entire spectrum was recorded for the higher DMPO concentration series (1.7–6.0 M in water and neat DMPO). The EPR measurements were started approximately 2–3 min after irradiation and repeated 10 times every 1 min (for a single line) or 7 times every 1.5 min (for the entire spectrum).

EPR data acquisition was controlled by the WIN-RAD ESR Data Analyzer System (Radical Research, Inc., Hino, Tokyo, Japan). EPR spectra were measured under the following conditions: microwave frequency: 9.45 GHz, microwave power: 2 mW, time constant: 0.01 s, field modulation frequency: 100 kHz, and field modulation width: 0.063 mT. The lower magnetic field: 336.2 mT for TEMPOL or 335.5 mT for DMPO-OH, field sweep width: +0.75 mT, sweep time: 15 s, data resolution: 1024 points for scanning single line. For scanning of the entire spectrum, the lower magnetic field: 326.7 mT, field sweep width: +10 mT, sweep time: 60 s, data resolution: 4096 points.

### 4.5. Preparation and EPR Measurement of Standard TEMPOL Solutions

Milli-Q water, 1.7 M, 2.3 M, 3.2 M, 4.8 M, and 6.0 M DMPO water solution, and neat DMPO liquid was used as the solvent to prepare 0.1 mM TEMPOL solutions. The center line of triplet of TEMPOL or whole triplet was scanned by X-band EPR with the same parameters as described above. The sample solutions/liquids were held in the PTFE tubing, and the sample volume occupying the sensitive volume in the cavity was the same for all samples measured. The EPR signal intensity of DMPO-OH or DMPO adducts in neat DMPO was converted to the radical concentration by comparing the signal intensity of corresponding TEMPOL solutions as a standard.

### 4.6. Estimation of EPR Signal Intensity by Peak Area

The acquired single line EPR spectra were analyzed using an in-house line fitting program (Appendix A), and the Gaussian line shape was fitted. Examples of line fitting results are shown in Figure 5. First, the highest and lowest peaks of experimentally observed spectra (Figure 5A,B) are automatically searched for relatively low-noise data or manually selected for noisy data, and then the initial estimation of the center and the peak-to-peak linewidth was obtained. Next, the center (H_0_) and peak-to-peak linewidth (H_pp_) of the calculated Gaussian line (Figure 5C,D) were adjusted to make the R^2^ value between the experimental spectrum and calculated Gaussian line maximum. Then, the height (I_pp_) of the Gaussian line was adjusted to minimize the difference between the experimental spectrum and calculated Gaussian line (Figure 5E,F). The signal height and line width of the best fitted Gaussian line were recorded, and EPR signal intensity, I_area_, was obtained as Equation (10).
I_area_ = I_pp_ × H_pp_^2^ × 1.03(10)

The 3-fold value of the estimated area of the 2nd line of DMPO-OH was used for the total spin amount of DMPO-OH in the sample solution. The 3-fold value of the estimated area of center line of TEMPOL was used for the total spin amount in the standard TEMPOL solution.

For whole-range EPR spectral data, digital values were integrated point-by-point to obtain an integral spectrum (Figure 6). An original differential EPR spectrum was corrected with 4096 digital data points (Figure 6A). The differential spectrum was simply integrated point-by-point and the integral spectrum was obtained (Figure 6B). The integrated spectra often have baseline distortion due to random broad noises. The distorted baseline profile with 17 adjustable points (solid black line with dots in Figure 6C) was manually drawn on the original integral spectrum (dark gray line in Figure 6C). Values between the 17 adjustable points were lineally interpolated. The estimated baseline was subtracted from the original integral spectrum and the corrected integral spectrum with an almost flat baseline (light gray line in Figure 6C) was obtained. The integral spectrum after the baseline correction (Figure 6D) was used for area estimation by second integration. Integration of spectrum and baseline correction processes were performed on Microsoft Excel 2010.

### 4.7. Correction of the Decay of EPR Signal

The natural decay rate, *k*_EXP_, of DMPO-radical-adduct was estimated from the slope of the semi-logarithmic plot of DMPO-radical-adduct versus time after irradiation. The concentration of the DMPO-radical-adduct at the end of X-ray irradiation (time = 0), C_0EXP_, was obtained by extrapolating the experimental decay curve of the DMPO-radical-adduct to time = 0. As the DMPO-radical-adducts decayed both after and during irradiation, C_0EXP_ is an underestimated value of the net amount of the DMPO-radical-adduct.

To obtain the net amount of DMPO-radical-adduct generated during X-ray irradiation, C_0NET_, the linear generation of DMPO-radical-adduct was corrected considering its first-order decay during X-ray irradiation, *k*_IRD_. Combination of a zero-order increasing with the rate C_0NET_/T_irrad_ and first-order decay with the decay rate *k*_IRD_ are simultaneously running during X-ray irradiation, where T_irrad_ is total irradiation time. The net amount of DMPO-radical-adduct generated during X-ray irradiation, C_0NET_, was estimated by Equation (11).
C_0NET_ = C_0EXP_ × T_irrad_ × *k*_IRD_/(1 − EXP(−*k*_IRD_ × T_irrad_))(11)

According to the previous paper [31], *k*_IRD_ = 3.05 × *k*_EXP_ was used for correcting the decay of DMPO-OH during irradiation. DMPO-OH is a nitroxyl radical that can be reduced by the H_2_O_2_-related reaction during X-ray irradiation [26]. The decay rate of DMPO-OH during irradiation is faster than that after irradiation due to the additional reduction of the nitroxyl radical, i.e., DMPO-OH. The yields of DMPO-OH should be linear with dose. Therefore, a suitable decay correction is necessary [31].

## 5. Conclusions

The first purpose of this study was to improve the quantification procedure for •OH based on EPR spin trapping using DMPO in high-concentration DMPO solutions, and the second purpose was to estimate the local linear-density (local concentration) of initial •OH. Both purposes were achieved in this study. Suitable decay correction of DMPO-OH adducts is necessary to accurately estimate the •OH yield. Baseline correction of the integral spectrum is necessary to accurately estimate the double integration value, i.e., area, to quantify the amount of spin. A DMPO concentration higher than 3.2 M may trap most (91%) of the X-ray-induced •OH generated by water radiolysis. The linear-density of markedly dense •OH induced by X-rays in water was estimated as 1168 μm^−1^, which can be converted to 0.86 nm as the intermolecular distance and 2.6 M as the local concentration.

## Figures and Tables

**Figure 1 molecules-27-00592-f001:**
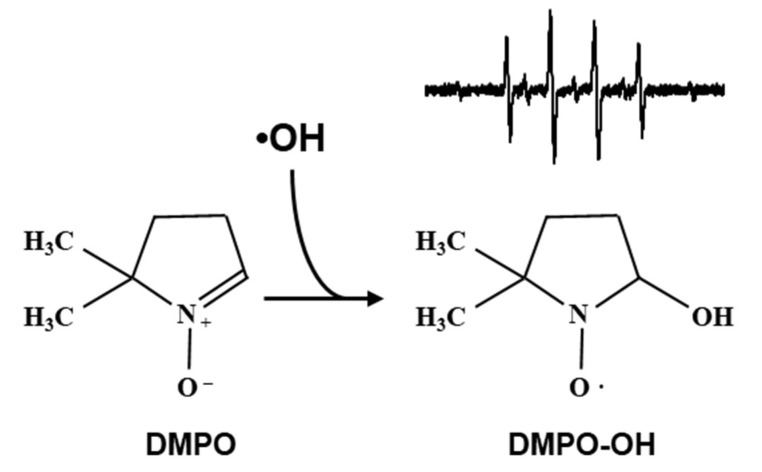
Spin-trapping of •OH by DMPO. DMPO, which is a diamagnetic and EPR silent molecule, can react with •OH and make an •OH adduct of DMPO (DMPO-OH), which has a relatively stable nitroxyl radical. This can be measured using EPR spectroscopy. A typical EPR spectrum of DMPO-OH, which has 4 lines with an intensity ratio of 1:2:2:1, is shown above the structure of DMPO-OH.

**Figure 2 molecules-27-00592-f002:**
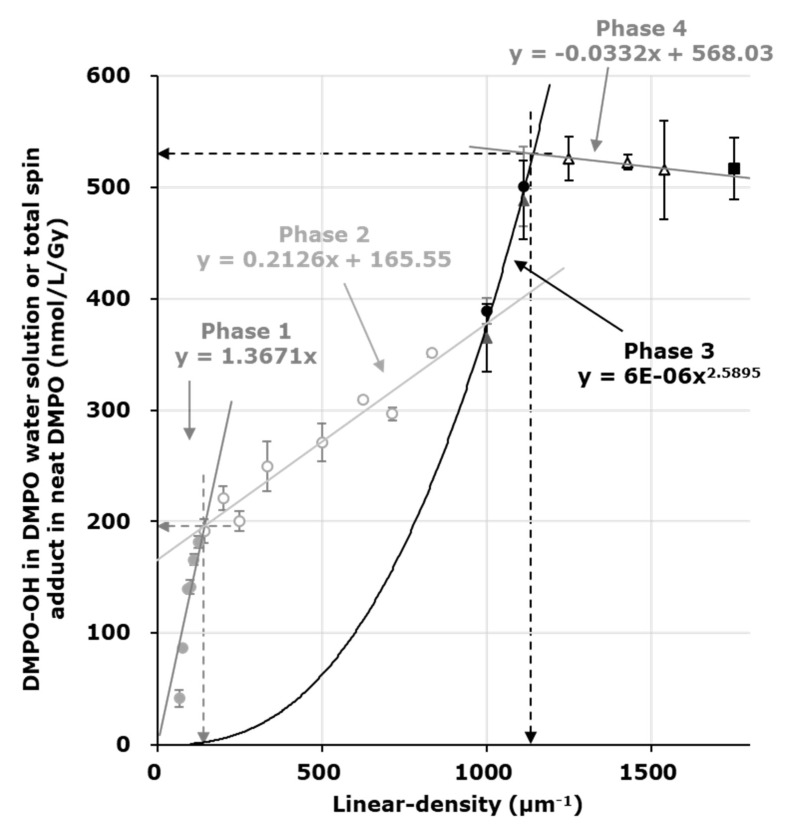
Profile of X-ray-induced DMPO-OH concentrations in the sample solutions plotted versus the linear-density of DMPO. The linear-density of relatively sparse •OH generation can be estimated from the inflection point of phase 1 (solid gray circles) and phase 2 (open circles) as 142 μm^−1^ (down gray dotted arrow), which was converted to 7.0 nm as the intermolecular distance and 4.8 mM as the local concentration. Data obtained using the higher concentration series of DMPO solutions (triangles and a square) were plotted over the data observed using the lower concentration of DMPO solutions. Using a higher concentration series of DMPO, the 4th phase was able to be observed. The linear-density of markedly dense •OH generation was estimated from the inflection point of phase 3 (gray circles and black triangles) and phase 4 (open triangles) as 1168 μm^−1^ (down black dotted arrow), which was converted to 0.86 nm as the intermolecular distance and 2.6 M as the local concentration. The value obtained for neat DMPO (black square) is just a reference and was not used to obtain phase 4. The marks and error bars indicate the average values and the standard deviations of 3–5 measurements, respectively. The regression line was fit to phases 1, 2, and 4, and that for phase 1 was through the origin. An exponential line was fit to Phase 3.

**Figure 3 molecules-27-00592-f003:**
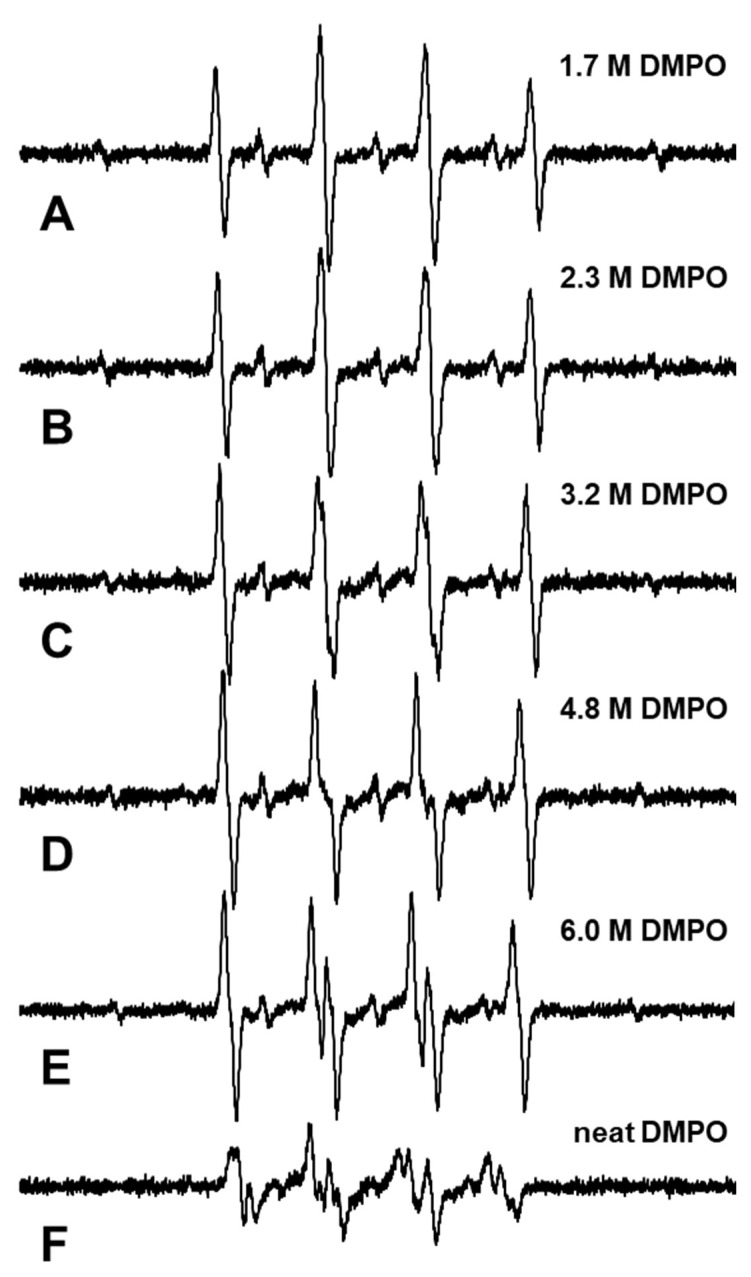
EPR spectra obtained in the higher concentration DMPO water solutions and neat DMPO after 32-Gy X-ray irradiation. (**A**) EPR spectrum obtained in the 1.7 M DMPO solution. Typical spectral shapes of DMPO-OH and DMPO-H were observed and distinguished. (**B**) EPR spectrum obtained in the 2.3 M DMPO solution. The linewidth was slightly broader than that in (**A**). (**C**) EPR spectrum obtained in the 3.2 M DMPO solution. (**D**) EPR spectrum obtained in the 4.8 M DMPO solution. (**E**) EPR spectrum obtained in the 6.0 M DMPO solution. (**F**) EPR spectrum obtained in the neat (99%, 8.8 M) DMPO liquid.

**Figure 4 molecules-27-00592-f004:**
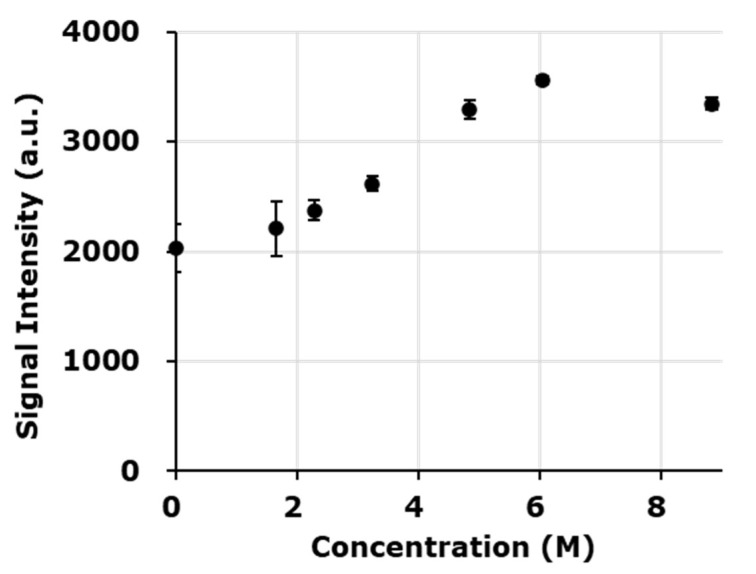
Comparison of the X-band EPR signal intensities of 0.1 mM TEMPOL solutions containing different concentrations of DMPO/water. The same volumes of samples in PTFE tubing were measured at the same EPR conditions, except coupling and phase adjustment. The resonance conditions, i.e., frequency, coupling, and phase, were adjusted before each measurement. The EPR conditions are described in the text. Marks and error bars indicate averages and standard deviations of more than 3 experiments. The 8.9 M DMPO is 99% neat DMPO liquid.

**Figure 5 molecules-27-00592-f005:**
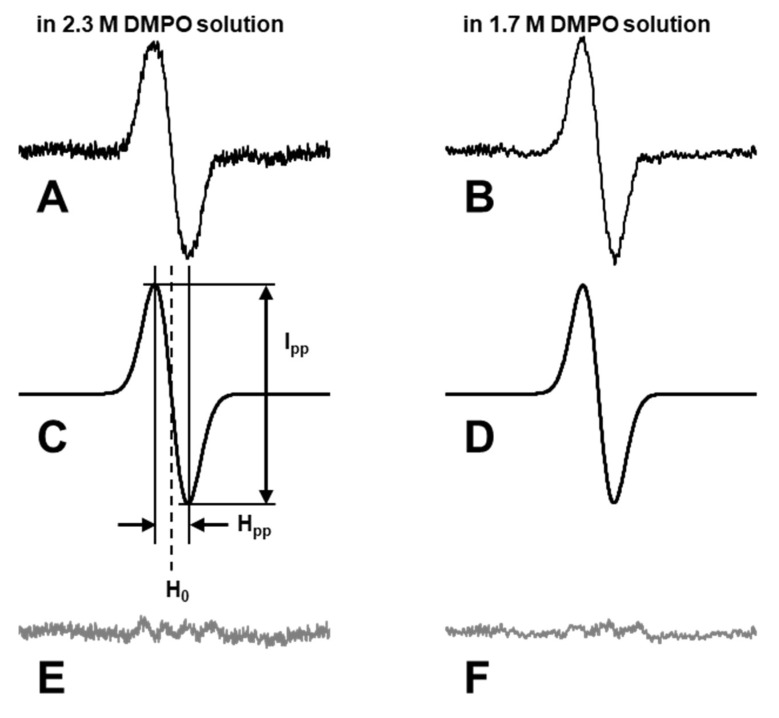
Examples of line fitting. The 2nd line from the lower field of the 4-line EPR spectrum of DMPO-OH caused in (**A**) 2.3 M and (**B**) 1.7 M DMPO water solution irradiated with 32 Gy of X-rays. (**C**,**D**) Best fitted Gaussian lines calculated for (**A**,**B**), respectively. I_pp_ is the peak-to-peak signal height. H_pp_ is the peak-to-peak linewidth. H_0_ is the center of the Gaussian line. (**E**,**F**) Residues of subtracting the calculated Gaussian line from the experimental spectrum.

**Figure 6 molecules-27-00592-f006:**
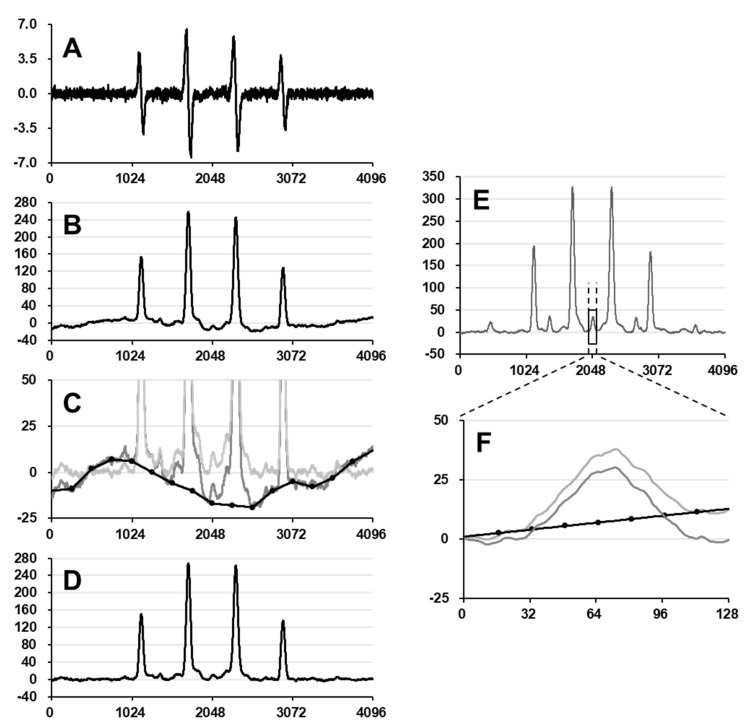
An example of area estimation for whole-range EPR spectrum. (**A**) An original differential spectrum of DMPO-OH observed in 1.7 M DMPO water solution irradiated with 32 Gy of X-rays. Digital data were corrected as 4096 data points. (**B**) The integral spectrum of (**A**) was obtained by simple point-by-point integration. Baseline distortion can be visualized. (**C**) The distorted baseline manually drawn on the integrated spectrum (solid line with dots) was subtracted from the integrated spectrum. (**D**) Integral spectrum after the baseline correction. (**E**) A baseline-corrected integral EPR spectrum, which included components of both DMPO-OH and DMPO-H, obtained in a 1.7 M DMPO water solution irradiated with 32 Gy of X-rays. (**F**) The center peak of DMPO-H extracted from (**E**), and its baseline correction. To estimate the contribution of DMPO-OH to the spectrum shown in (**E**), a 6-fold value of the area of (**F**) was subtracted from the total area of (**E**). The X- and *y*-axis are data points and absolute EPR signal intensity, respectively.

**Figure 7 molecules-27-00592-f007:**
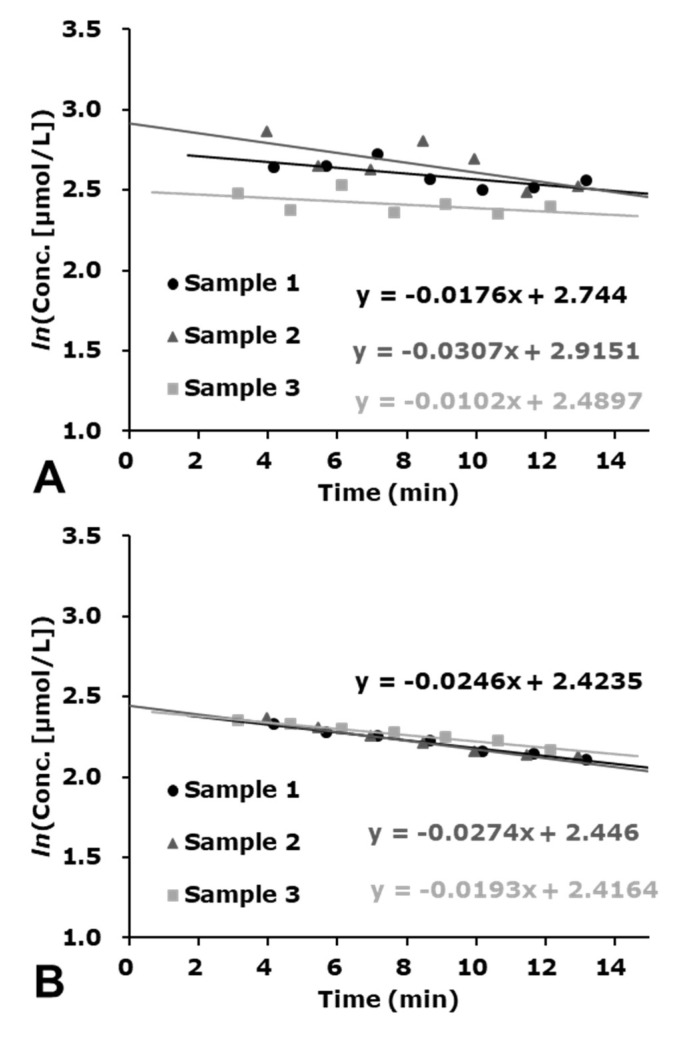
Comparison of data processed with or without baseline correction of the integral spectrum. (**A**) Decay profiles of the total spin-adduct caused in neat DMPO liquid irradiated with 32 Gy of X-rays were observed when the yield of the total spin concentration was estimated without baseline correction. (**B**) Data reproducibility was markedly improved by baseline correction of the integral spectrum.

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
