# Peer review of "Estimation of the Local Concentration of the Markedly Dense Hydroxyl Radical Generation Induced by X-rays in Water"

_molecules, 2022, doi:10.3390/molecules27030592_

Round 1

Reviewer 1 Report

The authors have made efforts to answer all the reviewers comments as thoroughly as possible.   However, some efforts on presenting a more straightforward presentation of the results would be helpful to the reader.

There are some points that are still not clear with their model of linear density.   At this time the authors may not be able to deal with more complex models but they need to consider the following in this work or in their subsequent efforts.

  1. It is true that the formation of radicals by x-radiation is inhomogeneous and therefore a model that each ray deposites the energy in a line may be a better approximation than homogeneous. However, it is an extreme limit and not truly representative of the track structure.   Thus the authors would be aided by a comparison to track models that are quite accurate in simulating the deposition of energy and radical formation in water systems.

  1. Next at the highest DMPO concentrations the authors suggest there are about 3 waters per DMPO molecule.   The molecular weight of DMPO is 113 and that of 3 waters is 18X3 = 54.   So 2/3 of the energy will be deposited in the DMPO.   The ionized waters may also behave differently as there cation rations initially form can charge transfer to the DMPO and the DMPO cation radical would likely deprotonate to form a carbon center radical species which could add to a second DMPO double bond.  Thus the analysis of the radicals species mass spectra would help to identify the species giving rise to the EPR signals.  Owing to the relative low concentration of the radicals this may be an impractical task but it should alert the authors they their system may be more complex than they are accounting for.

Author Response

Our responses to the reviewer's comments are provided as an attachment.

Reviewer 2 Report

The authors have made a numberof changes in the manuscript based on the comments of the reviewers. The changes answered some of the questions from the original reviews. Unfortunately the changes also made the paper significantly harder to read and more poorly organized; some important controls are still lacking and some major flaws in the researtch and its design are clearly revealed. 

  1. This work really depends, for both correctness and impact, on having accurate dosimetry of the samples. That is, how much X-ray energy is actually absorbed by the water, that is clear from the title of the manuscript. Dosimetry is an issue here because X-rays deposit energy in the water and in the solutes like DMPO dissolved int he water. When DMPO becomes a significant fraction of the solution (here up to 1 molecule in 3), dosimetry is an important concern. The X-ray absorption is different for different compounds, like water and DMPO. The manuscript reports absorbed X-ray doses for the total solutions but not for the different components. In addition, doses are reported in Gy, which is an appropriate absorbed dose unit. But the response to reviewers and the description added in line 411 that the absorbed dose can be measered by an ionization chanber next to the sample is incorrect and suggests an ignorance of a key aspect of this study. It is not simple do do the dosimetry properly but it is crucial to this work and needs the involvement of a professional.
  2. The composition of the samples is not reported accurately or in a transparent manner. It is important because the radiation chemistry of water and the spin trapping yields are dependent on relatively minor amounts of dissolved oxygen and dissolved carbon dioxide in the form of carbonate and bicarbonate ions. Such species react, some of their radical products can be spin trapped and are difficult or impossible to distinguish from the spin adducts measured in this study. Oxygen is important because it can produce DMPO-OH and because it can be depleted during irradiation, so that yields are not necessarily proportional to dose as claimed in the manuscript. Oxygen tends to be much more soluble in organic liquids than in water, so the conccentration of oxygen can be quite different in samples of different composition. It is not adequate to assume that aerated samples are all the same. Similarly, carbon dioxide is readily absorbed by water, altering pH and can build up to high levels. It is not adequate to assume some equilibrium especially in a series of samples of different compositions. There are standard analytical methods for both oxygen and carbon dioxide measurements, but these important controls were not applied.
  3. Uncertainties are reported for some quantities and some error bars are given, but error propagation is not carried to important results, like slopes and yields in Fig. 5, for example. 

The manuscript continues to ignore the important and highly-relevant work on initial yields and reactions from picosecond and sub-picosecond pulse radiolysis measurements which would add to the rlevance and significance of this manuscript.

The comment made in line 45 that for photon irradiation (which includes X-rays), "spurs are scattered and well separated " would seem to negate the whole purpose and conclusion of this manuscript.

The organization and clarity of the manuscript has become worse. It seems that the paper is now organized to answer the reviewers than to describe the rationale and results of the experiment. The English need to be improved.

Figure 2, which is redundant with Fig. 5 is still in the manuscript. They should be consolidated.

In view of the major flaws that compromise the integrity of the conclusions and the lack of proper controls.

Author Response

(The authors gave the same response as above.)

Reviewer 3 Report

The authors answered all of the reviewers comments in a satisfactory manner therefore I recommend the paper for publication

Author Response

(The authors gave the same response as above.)

Round 2

Reviewer 1 Report

The authors have  made changes in the manuscript in accord with the reviewers recommendations.   

Author Response

Our response to reviewer is summarized in the attached file.

Reviewer 2 Report

This paper is scientifically unsound, poorly written, lacks proper experimental controls and should be rejected. The paper has already had at least two rounds of reviews before this and some minor rewriting has been done but the major flaws remain and previous reviewer comments about them are not even acknowledged. The paper remains poorly organized and some sections are repeated. 

1) There is a fatal flaw concerning the dosimetry. The paper reports the absorbed dose (inherent in the units of Gy) for samples whose absorption of radiation varies because the sample composition varies. The comment in the paper saying that an ionization chamber was used to measure absorbed dose is a mistake showing confusion between exposure to radiation and absorption of radiation. Without accurate dosimetry the rest of the paper loses meaning.

2) The paper claims to report on the yield of OH radicals from the irradiation of water. But results are derived from samples that contain large amounts of DMPO which are also absorbing radiation and being ionized. Lines 350-383 report that in the region where the remarkably high yields of OH radicals are seen, the amount of DMPO varies from 20%-67% of the solution. On lines 454-455, it is said that the frequency of ionization in water and DMPO is likely the same, which is reasonably accurate. Thus, in these samples, the radiation absorbed by water decreases nearly 3-fold because the amount of water decreases that much. Yet, the yield of spin trapped OH radicals in the samples is roughly constant and is claimed to approach the absolute quantity of OH radicals produced from irradiation of water. This contains a major logical error. Either: a) the decreasing amount of water and radiation absorbed by it are producing much more OH radicals to compensate for the lower absorption; or b) the radiation absorbed by the DMPO in the samples is producing OH radicals by reactions that do not appear and are not discussed in reactions 1-9. In either case, it is not possible to conclude much about the existence or quantity of remarkably dense OH production.

3) The changes in the spectra in Fig. 3 as the amount of DMPO in the sample is increased can have several interpretations, including the change from one radical adduct to another. Indeed, 3C seems to capture both radicals clearly as a mixture of the 4-line spectrum in 3A and the 6-line spectrum in 3E. This could be decided rather convincingly by a suggestion I made in an earlier review by the simple control experiment where a sample with high DMPO was irradiated, but then diluted to 3B for measurement. If the supposition that the sample dielectric properties altered the spectrum is correct, this control would have the 4-line spectrum of 3B. The need for this control has not even been acknowledged in the subsequent revisions.

4) Some minor parts of this paper are repeated in the text. The source and properties of DMPO should appear only once in the experimental section. They should not be repeated in lines 318-328; in lines 365-370, and again in 463-467.

Similarly, the experimental details about the quantitation of the EPR spectra is given in lines 300-310 and 341-364 and again in 515-542. These descriptions are different enough to raise serious questions about how spectra were quantified. In one place a custom computer program was used, in another Microsoft Excel was used. Which is correct?

5) The concentration of oxygen in the samples seems to have changed from the previous revision where it appeared that some were aerobic and others anaerobic. Now they all seem to be aerobic and there is a misguided attempt to measure the oxygen concentration using lithium phthalocyanine. One of the advantages of lithium phthalocyanine in vivo is that it measures the chemical activity of oxygen, not its concentration. (This difference is explained in every introductory physical chemistry textbooks.) But it means that if a solution is equilibrated with air, the lithium phthalocyanine will report the same activity of oxygen as in air. This is very useful for in vivo measurements where the amount of oxygen in tissues is depleted by metabolism or even irradiation. One expects and indeed the authors found the same activity of oxygen in all samples using the lithium phthalocyanine, lines 437-441. But the concentration of oxygen could be very different between aqueous samples and those with a high content of organic material like DMPO. Consequently, proper controls are needed to rule out the involvement of oxygen in the reactions in samples with high DMPO concentrations.

These are by no means the only problems with the manuscript. Many are described by various reviewers in earlier reviews. The major problems are not being addressed in revisions, so this manuscript needs to be rejected.

Author Response

Our response to reviewer is summarized in the attached file.

This manuscript is a resubmission of an earlier submission. The following is a list of the peer review reports and author responses from that submission.

Round 1

Reviewer 1 Report

  This work uses DMPO to trap x-ray induced OH radicals in water and measures the concentrations of DMPO-OH by ESR.  The work comes to an apparently erroneous conclusion that the local concentration of OH radical can be as high as 2.8 M  from what appears to be an incorrect analysis.  The authors give 32 Gray of x-rays (80 kev) to their samples.  Since the g value of OH radicals is well known for water (ref 13 and 19 for example) and the max yield of OH radical is ca. 0.5 micromol/J at very short times.  This makes the highest concentration possible in their samples as 0.5 umol/Jx32 J/kg or 16  umol/kg  which is 16 microMolar not 2.8 M.      

  However, the authors do not measure their results at short times since the radiation is done at 3.1 Gy/min and then the samples are moved to the EPR. Thus the trapped DMPO-OH radicals 15 min later are homogeneously dispersed and no high inhomogeneous local concentrations can be measured.   

  The authors apparently did not account for oxygen present which can lead to Haber-Weiss chemistry and additonal OH radical production.    

Reviewer 2 Report

The authors use a spin trapping method in combination with EPR detection to determine the linear density of radiation induced OH* species in water. Unfortunately, they miss the opportunity to explain in the introduction, why this is of interest. The applied EPR methodology and analysis is standard.

Reviewer 3 Report

This is an interesting paper addressing the spatial distribution of radiation damage products using spin trapping at high trap concentrations to intercept radicals that are otherwise not observed because of their rapid recombination. This paper could be a significant contribution to several long lines of research into the spatial distribution of radiolysis products, including: chemical scavenging of products by high concentrations of scavengers; product yields and reactions in pico- and femto-second pulse radiolysis; and EPR measurements of trapped free radical spatial distributions. All these lines of research go back into the 1970's and 1980's but are largely ignored in this paper. This robs the paper of its full context and significance and will lessen its impact if there is no discussion of how the approach taken here and the results of this study relate with and extend previous work on this long standing problem.

The question of the spatial distribution of radiation is relevant to many areas, most notably radiation damage of polymers and other material, radiation therapy in medicine, and the biological effects of radiation. They all depend strongly on the spatial distribution of radiation products which is not simple even for X-rays in water where part of the absorbed dose produces small spurs, but where the secondary electrons or delta-rays produce denser tracks of products. There are many reviews that cover this question, one example covering EPR approaches is Applications of EPR in Radiation Research, eds. A. Lund and M. Shiotani, Springer, Heidelberg, 2014. This manuscript needs to acknowledge such previous work in order to highlight its significance and its original contributions.

This paper is the latest in a series on this subject by these authors. It appears that they might expect readers to be fully familiar with these papers, because there is a noticeable absence of important details and controls missing from this work. They might be in the previous papers, but a general reader not completely familiar with the previous papers, will have doubts and confusion from even a passing mention of the details and controls in previous papers. These include:

  1. There is some confusion about the radiation dose. The unit of Gy is for absorbed dose, implying that the radiation exposures took into account the differences in the absorption of the solutions at the high concentrations of DMPO. However, it is also likely that the dose reported as 32 Gy really means the dose that would have been absorbed by tissue equivalent, Fricke dosimeter, or whatever was used in calibrating the X-ray unit. So that as the solution changed from mostly water to half or more of the volume being DMPO, the absorbed  dose would change so that the radical concentration would not accurately reflect the yield at constant dose. The paper needs to disclose whether the reported dose actually represents the dose or the exposure, either would be acceptable, but transparency is needed here.
  2. Important details are missing about the samples. Reactions 4, 6 and 7 involve H+ and O2, making the concentrations of those reactants in the samples relevant. So wat was the pH of the buffered solutions and the milli-Q water used? Were the samples degassed? Did any CO2 dissolve in the samples to produce carbonate and bicarbonate ions which can also react with solvated electrons? Such details need to be disclosed so that the experiments are adequately defined and revealed.
  3. In addition to the chemical reactions of radiolysis products discussed in the paper and in point 2 above, DMPO and its spin trap products also have a rich chemistry that led the in vivo spin trappers to turn to other spin traps. In particular, the superoxide and hydroperoxyl adducts can react further to produce the hydroxyl radical adduct, confusing the quantitation of hydroxyl radicals if superoxide or hydroperoxy radicals are produced by reactions 6 or 7. Thus it is important to reveal in this paper whether the yield of the hydroxyl radical adduct is linear with dose at doses up to the 32 Gy used here. In these rather pure solutions, superoxide and other fairly stable radicals have a long time to react both with DMPO and with radiolysis products and products of products during radiolysis and the time before measurement. So, at a minimum, the linearity of yield with dose is a very necessary and pertinent control.
  4. The concept of linear density is poorly explained in this paper although it seems to be a highly quantitative and well-defined quantity. More explanation is needed along with how it depends on concentration, lifetime, cross-sectional area and diffusion constant.

There are also a few points that need more justification or correction.

  1. I have some problems with the analysis shown in Fig 3 and 7 involving the two black points for Phase 3. One of them lies exactly on the Phase 2 line. Why is it not part of the Phase 2 kinetics? One can argue that it shows no more spin adduct than one would expect from phase 2; and that the other black point is only 100 nmol/L/Gy larger than expected from phase 2. Why instead of these values of 0 and 100 does the phase 3 curve indicate values near 400 and 500? It is difficult to imagine that the increase in DMPO density suddenly turned off the chemical reactions operative in Phases 1 and 2?
  2. Similarly, Fig. 7 shows the sample of neat DMPO used to determine Phase 4. Why is the yield of spin adducts in a sample free of water relevant to the radiolysis of water? If it is, then that one sample is also relevant to the radiolysis of every other liquid that DMPO is miscible with; and indicates that they all have the same yield of hydroxyl radical, even for compounds containing no oxygen atoms.
  3. The dosimetry of solutions in the Phase 4 region needs careful consideration and justification. DMPO is a major fraction of the volume of these sample, sometimes greater than 50% of the volume. It is not clear if the X-rays absorbed by the DMPO is relevant to the radiolysis of water. It is also possible that DMPO in contact with a highly excited water molecule could directly react with it and intercept the sequence in reaction 5.
  4. If one excludes the point for neat DMPO in Figure 7, then all the points indicated as Phase 3 and 4 lie on a straight line passing through the origin? This would seem to be a 'null-hypothesis' that requires consideration.
  5. A minor point is that the statement on line 193 that  the DMPO-H adduct has an intensity '6-fold' that of its center line. I would expect 3-fold for this adduct. Also if only the amount of the DMPO-OH radical is desired, it is possible to integrate only one of its lines (with multiplication by an appropriate statistical factor of 2- or 3-fold) over a very limited baseline, which can be done with much better precision than integration of the entire spectrum and baseline.
  6. The statement is made near line 150 that changes in the solvent composition change its dielectric properties and hence the EPR spectrum of the DMPO adduct with hydroxyl radicals, accounting for the change in spectra in Fig. 4A-D. I do not find this argument convincing on spectral grounds. The spin adduct in 4A and B has almost identical hyperfine couplings for the proton and the nitrogen in the adduct. But the line intensities and the inner line shapes in 4D require that the proton and nitrogen have slightly different couplings. However, the shape of the the inner lines in 4C is incompatible with a single spin adduct of DMPO, but is consistent with a mixture (or linear combination of spectra) of two spin adducts, the one in 4D and the one in 4A and B. The paper describes homogeneous solutions which would have uniform dielectric properties and only a single spectral form. So, I doubt that 4B and 4D show the same spin adduct; but there is a very simple control experiment that needs to be reported in this paper. The samples in 4B, C and D need to be repeated and after irradiation, either unirradiated DMPO or unirradiated buffer must be mixed with the samples. For example the irradiated 2.3 M DMPO sample would be mixed with DMPO to a final concentration of 3.2M or 4.8 M and the EPR spectra measured. Then it would be clear if the spectral shapes are a property of the sample at the time of measurement or at the time of irradiation. Without such a proper control experiment, any conclusions drawn from high concentration samples are open to dispute.

The paper also would benefit by major reorganization. In particular figure 3 duplicates what is in Figure 7. Also, Fig. 6, 8 and 10 should be combined. The English in the paper has problems, particularly in the introduction and discussion. The grammar and syntax are good, but the problem is with vocabulary where the word has several possible meanings. There are several cases where the wrong word is used. That wrong word shares several meanings  with words that would be correct. But it lacks the meaning that is needed. One example is 'locations' in line 55. One might use 'locales', 'environs', 'regions', 'origins'; but 'locations' suggest that one can give the coordinates in advance and that they do not change. This unexpected misuse of words made these sections of the paper unusually difficult to read quickly.

Significant revision of this paper would greatly increase its interest and impact for the general reader.

Reviewer 4 Report

In this paper the authors estimate the amount of hydroxyl generated in water by X-ray radiation. Overall it is a well presented work using known methodology of spin trapping and EPR spectroscopy. However, although EPR spectrum of DMPO/•OH spin adduct has a distinct line-shape this doesn't exclude the possibility of generating DMPO/•OOH which spontaneously converts to DMPO/•OH. In order to be sure, one must perform a control experiment with either DMSO which is a scavenger for •OH radicals or superoxide dismutase which is a scavenger for •OOH radicals.

Therefore I recommend to publish the paper after the authors performed the control experiment and exclude the possibility of generating superoxide radicals as well.

Attached is a protocol for performing the control experiment.
